# Biocontrol Activity of Volatile Organic Compounds Emitted from *Bacillus paralicheniformis* 2-12 Against *Fusarium oxysporum* Associated with *Astragalus membranaceus* Root Rot

**DOI:** 10.3390/microorganisms13081782

**Published:** 2025-07-31

**Authors:** Yan Wang, Jiaqi Yuan, Rui Zhao, Shengnan Yuan, Yaxin Su, Wenhui Jiao, Xinyu Huo, Meiqin Wang, Weixin Fan, Chunwei Wang

**Affiliations:** 1Shanxi Key Laboratory of Bioagent Utilization and Eco-Pesticide Innovation, College of Plant Protection, Shanxi Agricultural University, Jinzhong 030801, China; yan314319@163.com (Y.W.); jiaqi000605@163.com (J.Y.); 15535986900@163.com (R.Z.); y18862772567@163.com (S.Y.); suyaxin010518@163.com (Y.S.); 15582415270@163.com (W.J.); 19373693894@163.com (X.H.); sxndwmq1973@163.com (M.W.); 2Experiment Teaching Center, Shanxi Agricultural University, Jinzhong 030801, China; fwxshishei@163.com

**Keywords:** *Bacillus paralicheniformis*, volatile organic compounds, *Fusarium oxysporum*, biocontrol activity, *Astragalus membranaceus* root rot

## Abstract

Root rot, mainly caused by *Fusarium oxysporum*, is one of the most destructive diseases and leads to significant economic loss of *Astragalus membranaceus*. To develop an effective strategy for the management of this serious disease, a bacterial strain 2-12 was screened from *A. membranaceus* rhizosphere soil and identified as *Bacillus paralicheniformis* based on the phylogenetic analyses of gyrase subunit B gene (*gyrB*) and RNA polymerase gene (*rpoB*) sequences. Interestingly, the volatile organic compounds (VOCs) produced by *B. paralicheniformis* 2-12 exhibited potent antifungal activities against *F. oxysporum*, as well as fifteen other plant pathogens. Under scanning electron microscopy observation, hyphae treated with the VOCs exhibited abnormal variation such as distortion, twist, and vesiculation, leading to distinctive protoplasm shrinkage. After treatment with *B. paralicheniformis* 2-12 VOCs, the lesion diameter and disease incidence both reduced significantly compared to control (*p* < 0.05), thus demonstrating prominent biological efficiency. Moreover, *B. paralicheniformis* 2-12 VOCs were composed of 17 VOCs, including 9 alkanes, 3 alcohols, 3 acids and esters, 1 aromatic compound, and 1 alkyne compound. A total of 1945 DEGs, including 1001 up-regulated and 944 down-regulated genes, were screened via transcriptome analysis. These DEGs were mainly associated with membranes and membrane parts, amino acid metabolism, and lipid metabolism. The findings in this work strongly suggested that *B. paralicheniformis* 2-12 VOCs could be applied as a new candidate for the control of *A. membranaceus* root rot.

## 1. Introduction

*Astragalus membranaceus* (Fisch.) Bge var. *mongholicus* (Bge.) Hsiao (Leguminosae), also named Huangqi, belonging to the *Astragalus* genus in the Leguminosae family, is one of the most important Chinese medicinal herbs native to Northern China [1,2]. It has been well known for its various pharmacological activities such as antioxidant, anti-tumor, diuretic, vital-energy-tonifying, and tissue-generative effects for more than 2000 years [3,4]. Owing to an increasing demand, wild *A. membranaceus* sharply decreased and could not meet the needs in the market [5]. Recently, *A. membranaceus* has been widely artificially planted in Shanxi, Gansu, and Inner Mongolia in China [2,5]. However, the occurrence of various diseases has attracted more attention due to serious losses. Particularly, root rot, one of the most economically severe diseases, was considered as a key constraint to commercial production of *A. membranaceus*. Based on the survey, root rot can cause a severe yield loss of more than 30% [6]. Several *Fusarium* spp., such as *F. oxysporum*, *F. acuminatum*, *F. solani*, and *F. redolens*, have been reported to cause *A. membranaceus* root rot [2]. Nevertheless, *F. oxysporum* are found to be the dominant pathogen associated with root rot on *A. membranaceus* [7].

Due to their low cost, convenient operation, and high control effects, several chemical fungicides such as hymexazol, carbendazim, and azoxystrobin have always been used to prevent and control Fusarium root rot on Chinese medical herbs in the field [5,8]. Previous studies also indicated that carbendazim, prochloraz, and fludioxonil had strong inhibition effects on *Fusarium* spp. and could be considered as candidates for the management of *A. membranaceus* root rot [9]. However, the long-term and extensive use of chemicals may cause adverse problems such as antibiotic resistance, environmental pollution, and even harm to human health [10,11]. Thus, new sustainable measures are required to control *A. membranaceus* root rot in the field.

Biological control is usually regarded as one of the most promising methods that could exhibit broad antifungal spectrum, minimal residual toxicity, and environmental friendliness [10,12]. Biological control agents (BCAs) and their products were widely used to control harmful pathogens on crops [11]. The mechanisms of BCAs mainly include nutrient and space competition, induced resistance, and the production of antimicrobial substances [10]. Among the mechanisms, volatile organic compounds (VOCs) produced by BCAs, the important natural microbial antagonists, have attracted more great attention because of their renewability, biodegradability, and environmental safety [13]. Particularly, VOCs with a low molecular weight (<300 Da) have been proven to disperse easily in air and could play an important role in fungal inhibition and disease suppression without physical contact with pathogens or crops [2,14]. Previous studies determined that some BCAs, belonging to the genera of *Bacillus*, *Streptomyces*, and *Pseudomonas*, could produce antifungal VOCs and have been widely applied to control root rot in the field [15,16,17,18]. However, to our best knowledge, few reports were available on the biological effects of microbial VOCs on *A. membranaceus* root rot.

In this work, the aims are to screen and identify new BCAs, evaluate the inhibition activities of the VOCs to plant pathogens, investigate the biological efficiency of VOCs on *A. membranaceus* root rot, determine the compositions of the VOCs, and elucidate the response of VOCs to *F. oxysporum* at the molecular level. Our findings would be expected to provide a theoretical basis for the development of novel biofumigants in the future.

## 2. Materials and Methods

### 2.1. Plant Pathogens and Culture Media

*F. oxysporum* H1 was isolated from a diseased sample of *A. membranaceus* in Hunyuan county, Shanxi Province of China. Fifteen other plant pathogens (listed in Table 1) were provided by the plant pathology laboratory of Shanxi Agricultural University.

### 2.2. Isolation and Screening of Antagonistic Bacterial Strain

A total of 10 g of rhizosphere soil sample collected from a healthy plant was added to 100 mL of sterile distilled water and shaken at 28 °C and 220 r/min for 10 min. Antagonistic bacteria exhibiting good inhibition effects on *F. oxysporum* were isolated from the soil bacterial solution by the serial dilution method and further purified from single colonies on nutrient agar (NA) medium at 28 °C for 48 h. All the bacterial isolates were stored at 4 °C before use.

The antagonistic effect of the test isolate was determined using the dual culture method previously described by Alijani et al. [19] and Wang et al. [20]. A mycelial disk (6 mm diameter) from a freshly prepared 7-day-old culture was transferred to the center of a 20 mL potato dextrose agar (PDA) medium plate. A total of 100 μL of the bacterial inoculum with 1 × 10^6^ colony-forming units (CFUs)/mL was inoculated on another plate with 15 mL of NA medium. Then, the two plates were rapidly sealed with polyethylene (PE) film. A PDA medium plate containing pathogen and another plate containing NA medium alone were used as control. The experiment was repeated three times. After the VOC treatment, mycelial growth was evaluated in two cross directions each day for 7 days. The percentage of mycelial inhibition of the antagonistic bacterial isolate was calculated based on the formula: Percentage of mycelial inhibition (%) = (d_c_ − d_t_)/(d_c_ − d_0_) × 100, where d_c_ represents the colony diameter of the control, dt represents the colony diameter of the treatment, and d_0_ is the diameter of the mycelial plug [21]. To further determine the inhibition effects of the VOCs on mycelial growth, the wet and dry weights were also measured at 7 days after treatment.

### 2.3. Identification of Antagonistic Bacterial Isolate

The colony characteristics of the tested isolate were described according to Bergey’s Manual of Determinative Bacteriology. The physiological and biochemical characteristics of the tested isolate were determined according to Bergey’s Manual of Determinative Bacteriology. For accurate identification, molecular features, including gyrase subunit B gene (*gyrB*) and RNA polymerase gene (*rpoB*) gene sequences, were amplified using the polymerase chain reaction (PCR). PCR amplification for the target gene sequences was performed using the described program with minor modifications [22,23,24,25]. The PCR amplification products were resolved on 1% agarose gels and further sequenced by Sangon Biotech Co., Ltd. (Shanghai, China).

The target gene sequences were aligned with the reference sequences using BLAST-N tool in the GenBank database (https://novopro.cn/blast/tblastn.html, accessed on 9 June 2025). The known sequences with high similarity were downloaded and used to analyze the evolutionary trajectory of the tested isolate. Multiple sequence alignments were performed using the CLUSTAL X Ver. 2.0 program. Phylogenetic trees were constructed to reveal the phylogenetic relationship using MEGA 7.0.14 in accordance with the neighbor-joining statistical method.

### 2.4. Scanning Electron Microscopy (SEM) Analysis

After exposure to the VOCs for 5 days, fungal hyphae taken from the inhibition areas were used to reveal the external morphological changes of *F. solani* and *F. oxysporum*. Samples without VOC treatments were used as control. The fungal hyphae were fixed with 1.5% glutaraldehyde and 1% osmium tetroxide [20]. Then, the tested samples were sputter-coated with gold palladium and observed using a JSM-6490LV scanning electron microscope (SEM, JEOL Company, Tokyo, Japan).

### 2.5. VOC Activities of Antagonistic Bacterial Isolate Against Other Plant Pathogenic Fungi

To confirm the effects of VOCs from antagonistic bacterial isolates on plant pathogenic fungi, the inhibition activities of antagonistic bacterial isolates against fifteen plant pathogenic fungi were measured using the dual culture method as described in Section 2.2. Percentage of mycelial inhibition was also calculated using the formula described above. Each experiment was conducted in five replicates.

### 2.6. Biological Efficiency of the VOCs on A. membranaceus Root Rot

To confirm biological potential, the effects of VOCs emitted from antagonistic bacterial isolates on *A. membranaceus* root rot were assessed by a previous method with slight modifications [20,26]. The tested isolate was placed on an NA plate and incubated at 23 °C for 36 h. A single colony was transferred into 20 mL of sterile LB broth and shaken at 28 °C and 180 r min^−1^ for 15 h. Next, 100 μL of the cultural broth was placed on the surface of the LA plate. Five *A. membranaceus* roots were wounded with a sterilized needle (2 mm in diameter and 3 mm in depth) and dripped with 20 μL of conidia suspension (10^6^ conidia/mL). Subsequently, the LA plate and inoculated *A. membranaceus* roots were placed into a sealed container, with high relative humidity maintained (approximately 90%). A total of 100 μL of sterile water placed on the LA plate was used as control. After incubation at 25 °C for 7 and 14 d, the biological efficiency was evaluated based on the disease incidence and lesion diameter [20].

### 2.7. Extraction and Analysis of VOCs from Antagonistic Bacterial Strain

The tested strain was transferred into Luria–Bertani (LB) medium for 14 h at 25 °C. After addition of 5 mL of Luria–Bertani agar (LA) medium containing 5 mg/L of heptyl acetate, a solid-phase micro-extraction (SPME) tube was placed at a 45-degree angle to increase the inoculation area [27,28]. Next, 100 μL of bacterial suspension (10^6^ colony-forming units (CFUs)/mL) was added onto the surface of the LA medium in the SPME tube and then incubated at 25 °C for 1, 3, 5, and 7 days. The tube containing only LA medium was used as control. For VOC extraction, SPME fiber (75 μm CAR/PDMS) was inserted into the tube and exposed to the headspace at 40 °C for 30 min.

The extracted analytes were further separated and detected using gas chromatography–mass spectrometry (GC-MS). The GC-MS system consisted of a Trace-1300 gas chromatograph, equipped with a Trace-ISQ quadruple mass spectrometer (Thermo Fisher Scientific, Waltham, MA, USA). A TR-5MS capillary column (30 m × 0.25 mm) with 0.25 μm film thickness (Thermo Fisher Scientific, Waltham, MA, USA) was used for separation of the analytes with a split–splitless injector. Helium (99.999% purity) was used as the carrier gas at a flow rate of 1.0 mL/min. The starting temperature of the oven was 40 °C for 3 min, at a heating rate of 10 °C/min to 180 °C, and then ramped at 40 °C/min to 270 °C, held for 4 min. Mass spectrometry was performed in electron ionization mode at 70 eV. Mass spectra scan ranged from 40 to 400 *m*/*z*. Mass spectral data of the VOCs from bacterial samples were compared with those of the standard compounds in the National Institutes of Standards and Technology (NIST) databases. The compounds were further verified based on their direct match (SI), reverse match (RSI), and similarity in NIST database [26,29,30,31].

### 2.8. Transcriptome Sequencing

*F. oxysporum* mycelial plug (6 mm diameter) was placed onto PDA plate and treated with 100 μL of *B. paralicheniformis* 2-12 bacterial inoculum (1 × 10^6^ CFU/mL) using the dual culture method as described in Section 2.2. After incubation at 25 °C for 5 days, 1000 mg hyphae of fungal colony were collected from the control groups and the treatment groups, placed into a 1.5 mL tube, and immediately frozen in liquid nitrogen. The collected hyphae were stored at −80 °C before transcriptomic analysis.

Then, the total RNA of hyphae was extracted using TRIzol Reagent Invitrogen (Invitrogen Life Technologies, Carlsbad, CA, USA). The RNA integrity was measured using a 2100 Bioanalyzer (Agilent Technologies, Inc., Santa Clara, CA, USA) and quantified using a NanoDrop 2000 Spectrophotometer (NanoDrop Technologies, Wilmington, DE, USA). Total RNA was combined with Oligo (dT) magnetic beads to purify mRNA, and fragmentation buffer was used to fragment mRNA. A set of 6-mer primers was added, and reverse transcriptase was used to synthesize double-stranded cDNA using mRNA as the template. The ends of the double-stranded cDNA were flattened at the 3′ end, and an A tail was added. Next, the sequencing libraries were constructed using the TruSeq™ RNA sample preparation kit (San Diego, CA, USA). Subsequently, transcriptome sequencing was performed using the Illumina HiSeq×TEN platform [32]. Differential expression genes (DEGs) were screened by DESeq2 software (Version 1.42.0) based on the threshold of |log2 (fold change)| ≥ 1 and false discovery rate (FDR) < 0.05 [32,33]. Then, all the DEGs were mapped to Gene Ontology (GO) terms in the GO database [10]. Kyoto Encyclopedia of Genes and Genomes (KEGG) enrichment analysis was further conducted using KEGG Orthology Based Annotation System (KOBAS) software (Version 3.0) [34].

### 2.9. Data Analysis

Each experiment was repeated three times, and the data were expressed as mean ± standard deviation (SD). For normality assessment, the experimental data were evaluated using the Shapiro–Wilk test. Statistically significant difference was determined by one-way analysis of variance (ANOVA) via Duncan’s multiple range test at *p* < 0.05 using SPSS software version 19.0 (IBM Corp., New York, NY, USA).

## 3. Results

### 3.1. Isolation and Screening of Antagonistic Bacterial Isolate with Antifungal VOCs

To screen new biocontrol agents, a total of 559 strains were isolated from 111 soil samples. Moreover, 58 strains with strong antagonistic activities in vitro were screened using the dual culture method. As a result, a new bacterial isolate was designated as 2-12 and exhibited a potent percentage of mycelial inhibition of 60.41% at 7 days after VOC treatment (Figure 1a). The wet weight and dry weight of mycelia were further assessed in VOC treatment. Within 6 and 7 days, the wet weights were 0.0345 and 0.0203 g, while the dry weights were 0.0040 and 0.0124 g, which all exhibited significant differences compared to the control (*p* < 0.05, Figure 1b,c).

### 3.2. Identification of Strain 2-12

The colony of strain 2-12 is circular, slightly convex, opaque, and milky white on the NA plate (Figure 2a). Strain 2-12 is a Gram-positive, aerobic cell (Figure 2b). Under the SEM observation, strain 2-12 is rod-shaped and straight, measuring 0.5 to 0.8 μm × 1.5 to 3.0 μm (*n* = 50) (Figure 2c). In the phylogenetic trees based on *gyrB* and *rpoB* gene sequences, isolate 2-12 was grouped into the same clade with *B. paralicheniformis* strains and distantly separated from other *Bacillus* species (Figure 3). Isolate 2-12 was identified as *Bacillus paralicheniformis* based on morphological characteristics and phylogenetic analyses. Subsequently, *B. paralicheniformis* 2-12 was deposited in China Center for Type Culture Collection (CCTCC) under accession number CCTCC M 20232598.

### 3.3. SEM Analysis of F. oxysporum Hyphae

*F. oxysporum* hyphae were observed under SEM observation. Without the VOCs from *B. paralicheniformis* 2-12, the control hyphae exhibited normal cylindrical shapes and smooth surfaces (Figure 4a–c). However, *F. oxysporum* hyphae treated with the VOCs have many abnormal variations such as distortion, twist, and vesiculation (Figure 4d–f). In particular, the hyphae surfaces were significantly shriveled, thus leading to distinctive protoplasm shrinkage compared to the control groups (Figure 4).

### 3.4. VOC Activities of B. paralicheniformis 2-12 Against Fifteen Plant Pathogenic Fungi

Table 1 illustrates the VOC activities from *B. paralicheniformis* 2-12 against fifteen plant pathogenic fungi. The VOCs exhibited potent inhibitory effects on the fifteen fungi with percentages of mycelial inhibition of 31.88~94.96%. Most of the variables (except *F. graminearum* and *Botrytis cinerea*) presented high Shapiro–Wilk *p*-values (greater than 0.05), indicating the normal distribution of our variables.

### 3.5. Biocontrol Effects of B. paralicheniformis 2-12 VOCs on A. membranaceus Root Rot

Biocontrol effects of the VOCs were measured at 7 and 14 d after treatment (Figure 5). At 7 d after treatment, the disease incidence significantly decreased to 13.33%, while that of the control groups maintained at 100%. The lesion diameter on treatment groups reduced to 0.20 cm, while those of the control groups were 2.20 cm. At 14 d after treatment, the disease incidence still maintained low levels (23.33%), and the lesion diameter reduced to 0.78 cm. Moreover, the experimental data exhibited significant difference (*p* < 0.05) compared to the control groups (Figure 5).

### 3.6. Determination of B. paralicheniformis 2-12 VOCs

The VOCs emitted from *B. paralicheniformis* 2-12 were determined using the SPME-GC-MS method based on the retention time (RT), direct match (SI), and reverse match (RSI). As a result, 17 VOCs, including 9 alkanes, 3 alcohols, 3 acids and esters, 1 aromatic compound, and 1 alkyne, were found in *B. paralicheniformis* 2-12 (Table 2). Of these VOCs, alkanes were found to be the most diverse class and might be the most important component of antifungal VOCs.

### 3.7. Transcriptomic Analysis

#### 3.7.1. Quality Control of Data

The control groups and treatment groups were set with three biological replicates for each treatment. A total of 361 million pieces for raw reads and 360 million pieces for clean reads were obtained through RNA-seq analysis. The Q20 data of the control and treatment groups ranged from 99.54% to 99.58%, while the Q30 data ranged from 97.23% to 97.42%, thus indicating good sequencing quality for bioinformatic analysis.

#### 3.7.2. Analysis of Differentially Expressed Genes (DEGs)

Under *B. paralicheniformis* 2-12 VOC stress, the distribution of DEGs in *F. oxysporum* was intuitively visualized in a volcano plot (Figure 6a). A total of 1945 DEGs, including 1001 up-regulated and 944 down-regulated genes, were identified compared to the control groups based on the criteria of *p*-value < 0.05 and fold change ≥ 2. Next, a heatmap was used to reveal the expression levels for the tested samples. Consequently, three treatment samples were clustered in one clade and separated from the control samples (Figure 6b).

#### 3.7.3. GO and KEGG Annotation Analyses

GO annotation analysis showed that these DEGs were mainly classified into three main functional categories: biological processes, cellular components, and molecular functions. Of these categories, the most significantly enriched GO terms were related to catalytic activity, membranes, metabolic processes, and membrane parts (Figure 5b). The results of KEGG analysis showed that these DEGs were involved in metabolism, genetic information processing, environmental information processing, and cellular processes. Among the 20 KEGG pathways, carbohydrate metabolism, amino acid metabolism, and lipid metabolism were found to be the most important pathways, accounting for large percentages of 14.65%, 20.41%, and 20.41%, respectively (Figure 5b).

#### 3.7.4. GO and KEGG Enrichment Analyses

To illustrate the functions of the DEGs, GO and KEGG enrichment analyses were conducted in this work. The DEGs were classified into 170 GO terms, mainly including transmembrane transport, transmembrane transporter activity, transporter activity, and oxidoreductase activity (Figure 7a). The KEGG enrichment analysis showed that the DEGs were mostly enriched in tryptophan metabolism, glycine, serine and threonine metabolism, steroid biosynthesis, phenylalanine metabolism, and pentose and glucuronate interconversions (Figure 7b).

## 4. Discussion

In this study, a new bacterial isolate 2-12 exhibited a potent mycelial inhibition effect on *F. oxysporum*, and its VOCs were also found to present noticeable inhibition activities against *F. oxysporum*, as well as fifteen other plant pathogens. Isolate 2-12 was further identified as *Bacillus paralicheniformis*. Although *B. paralicheniformis* has been used to control various diseases such as root rot, anthracnose, leaf spot on Quinoa, pearl millet blast, and tomato wilt [35,36,37], this is the first study demonstrating the antifungal activity of VOCs produced by *B. paralicheniformis*. Therefore, this study will provide a promising candidate for the development of new biocontrol agents.

Numerous reports have demonstrated that producing VOCs from microorganisms is a promising biological strategy owing to the prominent advantages of VOCs, including easy volatilization, non-physical contact, low residual, and high inhibitory activity [2,10,11,38]. *Bacillus* spp. have been reported to present strong antifungal activities against plant pathogens and are increasingly applied as the important biological control agents [34]. It is proved that the VOCs produced from *B*. *subtilis*, *B. siamensis*, *B. velezensis*, and *B*. *aryabhattai* could exhibit excellent biocontrol potential to various fungal pathogens in the field [39,40,41,42]. In this work, a new biological control agent, *B. paralicheniformis* 2-12, was found to produce antifungal VOCs. The percentage of mycelial inhibition was up to 60.41%. Moreover, *B. paralicheniformis* 2-12 VOCs also exhibited potent inhibitory effects on the other fifteen plant pathogenic fungi, indicating significantly antifungal activity and broad-spectrum inhibition characteristics. To reveal the morphological variations, *F. oxysporum* hyphae was further observed under SEM after treatment with *B. paralicheniformis* 2-12 VOCs. SEM analysis demonstrated that, compared to the control groups, the hyphae were distinctly twisty, vesicular, and shriveled on the surface, which is in agreement with previous reports [2,33]. In this study, new evidence suggested that VOCs produced by *B. paralicheniformis* could destroy the hyphae structure, thus leading to the inhibition of *F. oxysporum* hyphal growth.

Several reports have determined that the composition of bacterial VOCs mainly included alcohols, alkenes, aldehydes, acids, esters, ketones, aromatic compounds, and sulfides [2,42,43]. For instance, Calvo et al. [41] highlighted that three VOCs (diacetyl, benzaldehyde, and isoamyl alcohol) released from *B. velezensis* strains were proved to have high inhibitory activities against various kinds of post-harvest fungal pathogens. Wang et al. [2] revealed that *B. siamensis* YJ15 could produce 8 VOCs, including 4 alcohols, 1 ketone, 1 ester, 1 acid, and 1 aromatic compound, with high antagonistic activity against *B. cinerea*. Song et al. [42] reported that *B. aryabhattai* AYG1023 could produce 6 VOCs, including 4 ketones, 1 alcohol, and 1 aldehyde. Similarly, in this work, the VOCs produced by *B. paralicheniformis* were categorized as 9 alkanes, 3 alcohols, 3 acids and esters, 1 aromatic compound, and 1 alkyne compound. Although ethyl 2-methylbutanoate, one of the VOCs produced by *B. siamensis* YJ15, has been proved to exhibit antifungal activity against *B. cinerea* [2], the antifungal activities of other VOCs still remain largely unknown. So, further studies are strongly recommended to evaluate the activities of each volatile compound and to help reveal the antifungal mechanism.

To reveal the influence of VOCs on *F. oxysporum* at the molecular level, transcriptomic analysis was performed to investigate differential gene expression. Under *B. paralicheniformis* 2-12 VOC stress, a total of 1945 DEGs were found compared to the control groups. GO annotation analysis indicated that many DEGs were involved in catalytic activity, membranes, metabolic processes, and membrane parts. Cell membranes are of importance for cell integrity and fungal survival [33]. This finding showed that membrane and membrane components might be the main targets of the VOCs, which is in good agreement with previous studies as described by Yue et al. [10] and Duan et al. [33]. Additionally, the main KEGG pathways, such as amino acid metabolism and lipid metabolism, were also found through KEGG annotation analysis. Nishida et al. [44] indicated that amino acid metabolism could serve as a major nutrient source for fungi and may represent a new target for antifungal mechanisms. Mccarthy and Walsh [45] also reported that amino acid transport and metabolism were identified as potential drug targets to treat fungal infections. Lipids could increase cell membrane stability and regulate its fluidity and permeability [39]. Chen et al. [46] noted that citral causes the disorder of lipid metabolism and further disrupts the biological membrane functions of *Alternaria alternata*. Thus, amino acid metabolism and lipid metabolism might play important roles in the growth inhibition of *F. oxysporum*.

## 5. Conclusions

This study identified a new BCA, *B. paralicheniformis* 2-12, that could produce antifungal VOCs. *B. paralicheniformis* 2-12 VOCs exhibited prominent mycelial inhibition activity against *F. oxysporum*, as well as fifteen other plant pathogens. SEM analysis demonstrated that the hyphae treated with the VOCs were distinctly twisty, vesicular, and shriveled on the surface. Additionally, *B. paralicheniformis* 2-12 VOCs also demonstrated potent biological efficiency. *B. paralicheniformis* 2-12 VOCs were composed of 17 VOCs, including 9 alkanes, 3 alcohols, 3 acids and esters, 1 aromatic compound, and 1 alkyne compound. Through transcriptomic analysis, a total of 1945 DEGs were obtained and might be closely related to membranes and membrane parts, amino acid metabolism, and lipid metabolism. Although more studies are required to explain how the VOCs affect the growth of *F. oxysporum*, the findings will give new insights into the suppression of *A. membranaceus* root rot in the field.

## Figures and Tables

**Figure 1 microorganisms-13-01782-f001:**
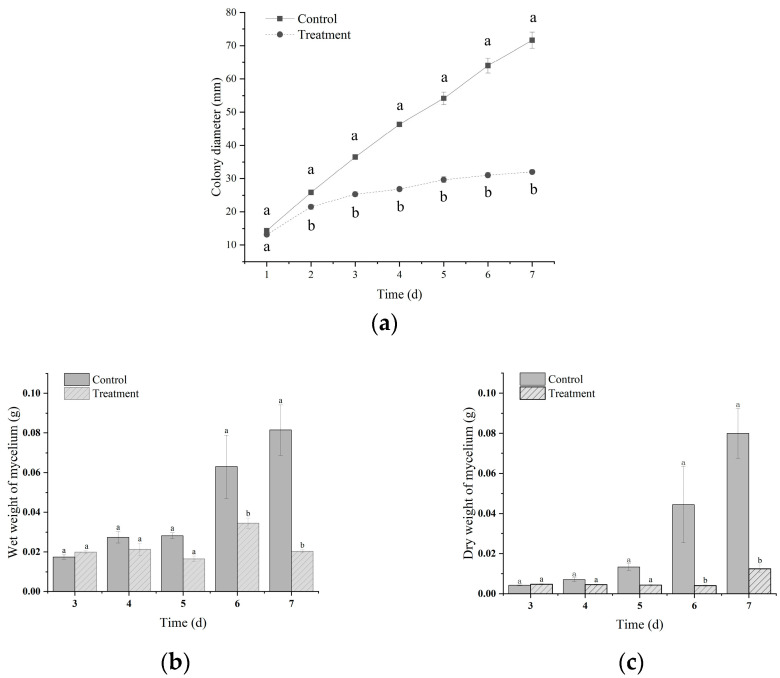
The antifungal activity of VOCs produced by isolate 2-12 against *Fusarium oxysporum*. (**a**) Mycelial growth; (**b**) mycelial wet weight; (**c**) mycelial dry weight. The same letter indicates no significant difference based on Duncan’s multiple range test.

**Figure 2 microorganisms-13-01782-f002:**
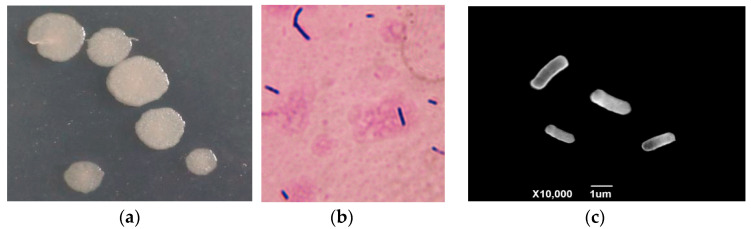
The morphological characteristics of isolate 2-12. (**a**) Colony on NA plate at 28 °C for 2 d; (**b**) Gram stain; (**c**) bacterial cell under transmission electron microscope.

**Figure 3 microorganisms-13-01782-f003:**
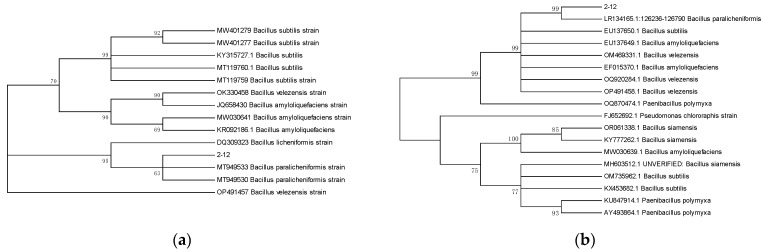
Phylogenetic tree based on *gyrB* (**a**) and *rpoB* (**b**) gene sequences.

**Figure 4 microorganisms-13-01782-f004:**
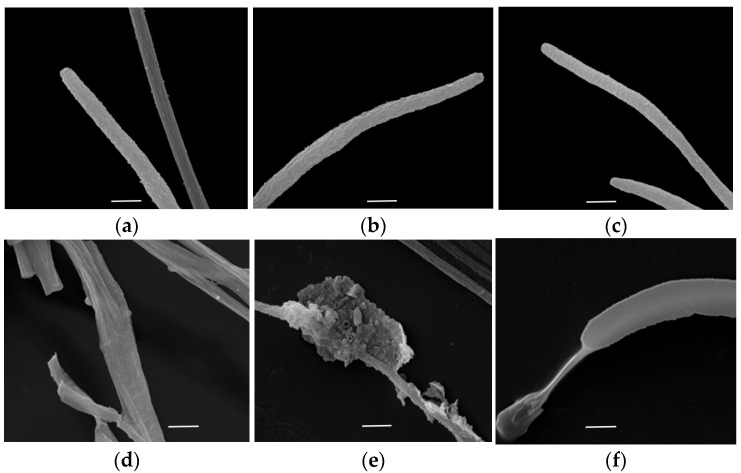
Scanning electron microscopy (SEM) analysis of *Fusarium oxysporum* hyphae. (**a**–**c**) hyphae treated without VOCs as controls; (**d**–**f**) hyphae treated with the VOCs. Scale bar = 5 μm.

**Figure 5 microorganisms-13-01782-f005:**
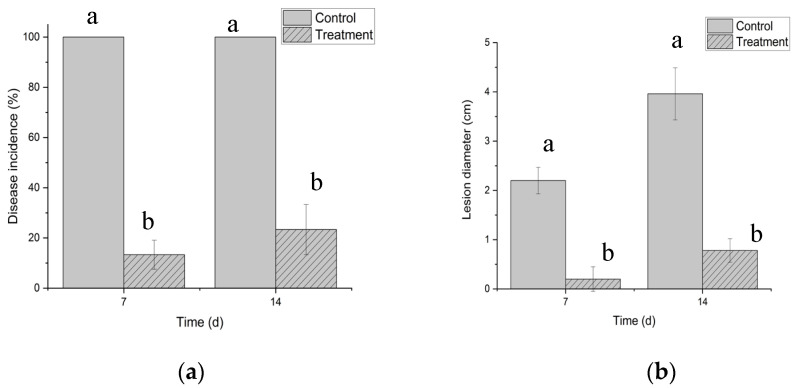
Biocontrol effects of *Bacillus paralicheniformis* 2-12 VOCs on *Astragalus membranaceus* root rot. (**a**) Disease incidence; (**b**) lesion diameter. The different letters mean significant difference based on Duncan’s multiple range test.

**Figure 6 microorganisms-13-01782-f006:**
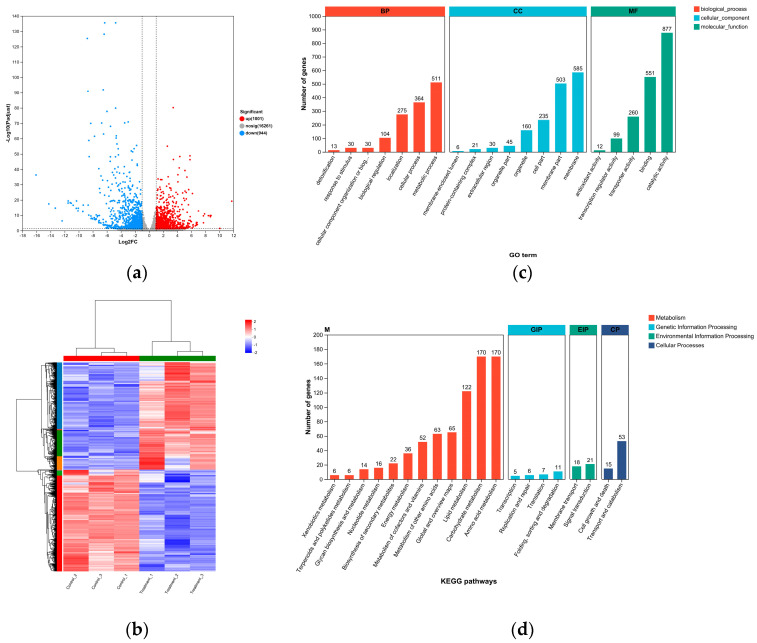
Transcriptomic analysis in *Fusarium oxysporum* under *Bacillus paralicheniformis* 2-12 VOC stress. (**a**) The scatter plot; (**b**) heatmap; (**c**) GO functional annotation analysis; (**d**) KEGG pathway annotation analysis.

**Figure 7 microorganisms-13-01782-f007:**
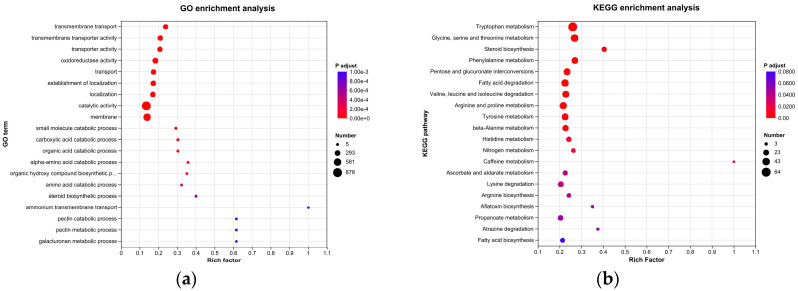
Enrichment analyses of DEGs. (**a**) GO enrichment analysis; (**b**) KEGG enrichment analysis.

**Table 1 microorganisms-13-01782-t001:** The antifungal activity of VOCs emitted from *Bacillus paralicheniformis* 2-12 against plant pathogenic fungi.

Fungi	Host	Percentage Mycelial Inhibition (%)	Shapiro–Wilk *p*
*Fusarium oxysporum*	*Phaseolus radiatus*	69.04 ± 0.63 de	0.244
*Fusarium solani*	*Astragalus membranaceus*	75.89 ± 1.93 bc	0.258
*Exserohilum turcicum*	*Zea mays*	77.22 ± 2.57 bc	0.550
*F. semitectum*	*Zea mays*	54.66 ± 2.00 h	0.843
*F. graminearum*	*Zea mays*	62.12 ± 2.89 fg	0.040
*Fusarium oxysporum*	*Musa paradisiaca*	63.06 ± 2.87 fg	0.420
*Botrytis cinerea*	*Cerasus pseudocerasus*	94.96 ± 2.28 a	0.042
*F. acuminatum*	*Astragalus membranaceus*	72.97 ± 6.15 cd	0.323
*F. avenaceum*	*Vitis vinifera*	31.88 ± 4.65 j	0.796
*F. equiseti*	*Astragalus membranaceus*	47.89 ± 4.45 i	0.453
*Fusarium graminearum*	*Triticum aestivum*	76.07 ± 2.03 bc	0.245
*Alternaria alternata*	*Chrysanthemum morifolium*	64.57 ± 2.88 ef	0.534
*Colletotrichum gloeosporioides*	*Capsicum annuum*	58.39 ± 3.47 gh	0.063
*F. oxysporum*	*Solanum melongena*	81.10 ± 0.32 b	0.828
*F. oxysporum*	*Cucumis sativus*	59.39 ± 5.00 fgh	0.328

The same letter indicates no significant difference based on Duncan’s multiple range test.

**Table 2 microorganisms-13-01782-t002:** The analysis of VOCs from *Bacillus paralicheniformis* 2-12 by SPME-GC-MS.

Compounds	Chemical Formula	MW ^1^	RT ^2^ (min)	SI ^3^	RSI ^4^
Ethyl 2-methylbutanoate	C_7_H_14_O_2_	130.18	6.09	693	748
Benzeneethanaminea-methyl	C_9_H_13_N	135.23	6.28	705	720
Propanoic acid, 3-amino-3-oxo-	C_3_H_5_NO_3_	103.08	6.31	603	952
N-Acetyl-L-alanine	C_5_H_9_NO_3_	131.13	6.32	592	981
2,4-Octadiyne	C_8_H_10_	106.17	6.92	744	848
Tridecane	C_13_H_28_	184.36	9.9	825	865
Undecane,5-methyl	C_12_H_26_	170.33	9.91	801	876
Decane,2,6-dimethyl	C_12_H_26_	170.33	10.66	620	901
3,5-dimethylheptan-3-ol	C_9_H_20_O	144.25	10.67	603	734
1-Butanol,2,2-dimethyl	C_6_H_14_O	102.17	10.67	670	775
2-Methylundecane	C_12_H_26_	170.33	13.32	835	864
3,3-Dimethylhexane	C_8_H_18_	114.23	14.00	742	786
2,3,4-Trimethylpentane	C_8_H_18_	114.23	14.14	555	764
Heptane,1-propoxy	C_10_H_22_O	158.28	14.36	580	671
2,5-Dimethyl-3,4-hexanediol	C_8_H_18_O_2_	146.23	14.36	603	691
Decane,2,9-dimethyl	C_12_H_26_	170.33	16.19	675	818
2,4,6-Trimethyloctane	C_11_H_24_	156.31	16.2	688	808

^1^ Retention time. ^2^ Molecular weight. ^3^ Direct match. ^4^ Reverse match.

## Data Availability

The original contributions presented in this study are included in the article. Further inquiries can be directed to the corresponding author.

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
