# Peer review of "Biocontrol Activity of Volatile Organic Compounds Emitted from Bacillus paralicheniformis 2-12 Against Fusarium oxysporum Associated with Astragalus membranaceus Root Rot"

_microorganisms, 2025, doi:10.3390/microorganisms13081782_

Round 1
Reviewer 1 Report
Comments and Suggestions for Authors
Dear Authors and Editor,
The manuscript presents an interesting and well-structured study on the biocontrol potential of volatile organic compounds produced by Bacillus paralicheniformis 2-12 against Fusarium oxysporum, the causal agent of root rot in Astragalus membranaceus. The work combines microbiological, microscopic, transcriptomic, and chemical analysis techniques, providing a valuable multidisciplinary approach. Its objective is clear and highly relevant within the current context of seeking sustainable alternatives to chemical fungicides. In my opinion, the manuscript does not present major scientific flaws and may be considered for publication. However, there are several grammatical and language issues throughout the text that must be corrected to improve its clarity. In some sections, the use of verb tenses and sentence structure makes the text difficult to understand. Below, I outline some of the errors I have identified:
Line 11: Replace lead by leads.
Line 51: Replace manage by management.
Line 55: Replace "root rot the field" by "root rot in the field"
Line 56: Replace "as a most promising method" by "as one of the most promising methods"
Line 63: Replace "with low-molecular-weight" by "with a low molecular weight"
Line 64: Replace "are proved to be easily disperse in air" by "have been proven to disperse easily in air"
Line 73: Replace "determine of the VOCs compositions" by "determine the compositions of the VOCs"
Line 78: Replace "were isolated" by "was isolated"
Line 90: Replace "and inoculate 100 μL" by "and 100 μL was inoculated"
Line 94: Replace "The mycelial growth were evaluated" by "The mycelial growth was evaluated"
Line 112: Replace "evolutionary rajectory" by "evolutionary trajectory"
Line 123: Replace "another plant pathogenic fungi" by "other plant pathogenic fungi"
Line 127: Replace "was also calculated the formula described above" by "was also calculated using the formula described above"
Line 152: Replace "by using chromatography– mass spectrometry (GC-MS)" by "by using Gas Chromatography – Mass Spectrometry (GC-MS) "
Line 203: Replace Botrytis cinerea by Fusarium oxysporum
Line 246: Replace "including 9 alkanes, 3 alcohols, 3 acids & esters, 1 aromatic compound and 1 alkyne compound" by "including 9 alkanes, 3 alcohols, 3 acids and esters, 1 aromatic compound, and 1 alkyne"
Line 287: Replace "its VOCs was also found to presented" by "its VOCs were also found to present"
Line 293: Replace "an outstanding candidate" by "a promising candidate"
Line 302: Replace "B. paralicheniformis 2-12 was found to produce antifungal VOCs, the percentage mycelial inhibition were up to..." by "B. paralicheniformis 2-12 was found to produce antifungal VOCs. The percentage of mycelial inhibition was up to..."
Line 338: Replace "used as an new target" by "used as a new target"
Some relevant comments:
Section 2.3, Line 107: I understand that when modifications are made to established protocols, they should be specified. Therefore, I suggest expanding the description of the methodology.
I also recommend modifying Figure 1a to better distinguish the curves of the treatment and the control.
Comments on the Quality of English LanguageComments regarding the quality of English were made above
Author Response
|
Comments 1: [Line 11: Replace lead by leads.] |
|
Response 1: [We have replaced and marked it in red] Thank you for pointing this out. We agree with this comment. Therefore, we have replaced it in red in Line 11, Paragraph 1, Page 1. |
|
Comments 2: [Line 51: Replace manage by management.] |
|
Response 2: [We have replaced and marked it in red] Thank you for pointing this out. We agree with this comment. Therefore, we have replaced it in red in Line 53, Paragraph 2, Page 2. |
|
Comments 3: [Line 55: Replace "root rot the field" by "root rot in the field"] |
|
Response 3: [We have replaced and marked it in red] Thank you for pointing this out. We agree with this comment. Therefore, we have replaced it in red in Line 56, Paragraph 2, Page 2. |
|
Comments 4: [Line 56: Replace "as a most promising method" by "as one of the most promising methods"] |
|
Response 4: [We have replaced and marked it in red] Thank you for pointing this out. We agree with this comment. Therefore, we have replaced it in red in Line 57, Paragraph 3, Page 2. |
|
Comments 5: [Line 63: Replace "with low-molecular-weight" by "with a low molecular weight"] |
|
Response 5: [We have replaced and marked it in red] Thank you for pointing this out. We agree with this comment. Therefore, we have replaced it in red in Line 65, Paragraph 3, Page 2. |
|
Comments 6: [Line 64: Replace "are proved to be easily disperse in air" by "have been proven to disperse easily in air"] |
|
Response 6: [We have replaced and marked it in red] Thank you for pointing this out. We agree with this comment. Therefore, we have replaced it in red in Line 65, Paragraph 3, Page 2. |
|
Comments 7: [Line 73: Replace "determine of the VOCs compositions" by "determine the compositions of the VOCs"] |
|
Response 7: [We have replaced and marked it in red] Thank you for pointing this out. We agree with this comment. Therefore, we have replaced it in red in Line 74, Paragraph 4, Page 2. |
|
Comments 8: [Line 78: Replace "were isolated" by "was isolated"] |
|
Response 8: [We have replaced and marked it in red] Thank you for pointing this out. We agree with this comment. Therefore, we have replaced it in red in Line 80, Paragraph 5, Page 2. |
|
Comments 9: [Line 90: Replace "and inoculate 100 μL" by "and 100 μL was inoculated"] |
|
Response 9: [We have replaced and marked it in red] Thank you for pointing this out. We agree with this comment. Therefore, we have replaced it in red in Line 92, Paragraph 1, Page 3. |
|
Comments 10: [Line 94: Replace "The mycelial growth were evaluated" by "The mycelial growth was evaluated"] |
|
Response 10: [We have replaced and marked it in red] Thank you for pointing this out. We agree with this comment. Therefore, we have replaced it in red in Line 97, Paragraph 1, Page 3. |
|
Comments 11: [Line 112: Replace "evolutionary rajectory" by "evolutionary trajectory"] |
|
Response 11: [We have replaced and marked it in red] Thank you for pointing this out. We agree with this comment. Therefore, we have replaced it in red in Line 117, Paragraph 3, Page 3. |
|
Comments 12: [Line 123: Replace "another plant pathogenic fungi" by "other plant pathogenic fungi"] |
|
Response 12: [We have replaced and marked it in red] Thank you for pointing this out. We agree with this comment. Therefore, we have replaced it in red in Line 128, Paragraph 5, Page 3. |
|
Comments 13: [Line 127: Replace "was also calculated the formula described above" by "was also calculated using the formula described above"] |
|
Response 13: [We have replaced and marked it in red] Thank you for pointing this out. We agree with this comment. Therefore, we have replaced it in red in Line 132, Paragraph 5, Page 3. |
|
Comments 14: [Line 152: Replace "by using chromatography– mass spectrometry (GC-MS)" by "by using Gas Chromatography – Mass Spectrometry (GC-MS) "] |
|
Response 14: [We have replaced and marked it in red] Thank you for pointing this out. We agree with this comment. Therefore, we have replaced it in red in Line 158, Paragraph 3, Page 4. |
|
Comments 15: [Line 203: Replace Botrytis cinerea by Fusarium oxysporum] |
|
Response 15: [We have replaced and marked it in red] Thank you for pointing this out. We agree with this comment. Therefore, we have replaced it in red in Line 210, Paragraph 4, Page 5. |
|
Comments 16: [Line 246: Replace "including 9 alkanes, 3 alcohols, 3 acids & esters, 1 aromatic compound and 1 alkyne compound" by "including 9 alkanes, 3 alcohols, 3 acids and esters, 1 aromatic compound, and 1 alkyne"] |
|
Response 16: [We have replaced and marked it in red] Thank you for pointing this out. We agree with this comment. Therefore, we have replaced it in red in Line 256, Paragraph 2, Page 8. |
|
Comments 17: [Line 287: Replace "its VOCs was also found to presented" by "its VOCs were also found to present"] |
|
Response 17: [We have replaced and marked it in red] Thank you for pointing this out. We agree with this comment. Therefore, we have replaced it in red in Line 306, Paragraph 2, Page 10. |
|
Comments 18: [Line 293: Replace "an outstanding candidate" by "a promising candidate"] |
|
Response 18: [We have replaced and marked it in red] Thank you for pointing this out. We agree with this comment. Therefore, we have replaced it in red in Line 311, Paragraph 2, Page 10. |
|
Comments 19: [Line 302: Replace "B. paralicheniformis 2-12 was found to produce antifungal VOCs, the percentage mycelial inhibition were up to..." by "B. paralicheniformis 2-12 was found to produce antifungal VOCs. The percentage of mycelial inhibition was up to..."] |
|
Response 19: [We have replaced and marked it in red] Thank you for pointing this out. We agree with this comment. Therefore, we have replaced it in red in Line 321, Paragraph 1, Page 11. |
|
Comments 20: [Line 338: Replace "used as an new target" by "used as a new target"] |
|
Response 20: [We have replaced and marked it in red] Thank you for pointing this out. We agree with this comment. Therefore, we have replaced it in red in Line 359, Paragraph 3, Page 11. |
|
Comments 21: [Section 2.3, Line 107: I understand that when modifications are made to established protocols, they should be specified. Therefore, I suggest expanding the description of the methodology.] |
|
Response 21: [We have revised and marked it in red] Thank you for pointing this out. We agree with this comment. Therefore, we have revised this part in red in Line 106, Paragraph 2, Page 3. |
|
Comments 22: [I also recommend modifying Figure 1a to better distinguish the curves of the treatment and the control.] |
|
Response 22: [We have modified Figure 1a. To better distinguish the curves, the solid line and dotted line were used for the treatment and the control, respectively.] Thank you for pointing this out. We agree with this comment. Therefore, we have modified Figure 1a in red in Line 209, Paragraph 3, Page 5. |
|
23. Response to Comments on the Quality of English Language |
|
Point 1: Comments regarding the quality of English were made above. |
|
Response 1: We are very sorry for the quality of English language. We have revised this manuscript carefully. |
|
24. Additional clarifications |
|
[We would like to thank you and the reviewers for the positive and valuable comments concerning our manuscript. We have checked it carefully. However, we are very sorry for our mistakes. Firstly, in Line 274 and Line 279, Paragraph 2, Page 9, Figure 6A and Figure 6A was changed to Figure 6a and Figure 6b. Secondly, we omitted the section of 3.7.3 GO and KEGG annotations analyses, and added this section in Line 279, Paragraph 2, Page 9. ] |

Reviewer 2 Report
Comments and Suggestions for Authors In this paper, the volatile organic compounds (VOCs) produced by a Bacillus species isolated from the rhizosphere of Astragalus membranaceus were explored as potential biological control agents (BCAs) alternatives to chemical fungicides. Fungal infection from Fusarium oxysporum was the chief concern of this paper but several other plant pathogens were mentioned as well. The efficacy of the Bacillus’ VOCs as a BCA were confirmed through microscopic observation of physical effect of the BCA treatment on F. oxysporum hyphae and by comparison of lesion formation on infected roots with and without VOC application. Gas chromatography and mass spectrometry were used to identify the 17 VOCs associated with this paper’s Bacillus strain. Transcriptomic data to identify differential gene expression in F. oxysporum after VOC treatment was also taken. In general, the results are convincing and support the conclusions the authors have provided. No additional experiments are necessary in my opinion. However, there are formatting and language issues which need to be addressed. A few examples are given here:- Some definitions for acronyms are missing throughout the paper or there are instances where acronyms are used before they are defined, particularly in the abstract.
- An instance of “Fusarium” is left unitalicized in line 49.
- Overall, the Methods section feels underdeveloped and should contain more technical information. In section 2.2 from the Methods, what the “antagonist” bacteria is is unclear. The sentences surrounding line 90 are unclear. The method of measurement used to supply data for the formula written in line 97 should be explained. It is unclear in section 2.6 if full plants are being used for this experiment or just roots severed from a plant. A more comprehensive explanation of statistical analysis should be given for the data collection than the one provided in section 2.9. statistics used need to be made clear in figure caption.
- In Figure 1a the differentiating markers (dot/box) between the control and treatment points are indistinguishable from one another.
- In line 237 it is reported that disease incidence “significantly decreased by 13.33%,” but the accompanying figure appears to show that disease incidence has actually decreased to 13.33%; this is most likely a grammatical error but the difference between 13.33% and 86.67% is major.
- I believe that Figure 5a is mislabeled on the 7 day treatment, both the control and treatment boxes are labeled “a” but I believe that they are actually statistically different from one another based on the rest of Figure 5.
No new experiments are recommended. With some editing this paper will be far more easy to read.
Comments on the Quality of English LanguageThe manuscript is difficult to follow in many places and would greatly benefit from some editing for clarity. These issues made the paper more difficult to follow during the review process and this will likely be an issue for readers.
Author Response
|
· Comments 1: [Some definitions for acronyms are missing throughout the paper or there are instances where acronyms are used before they are defined, particularly in the abstract.] |
|
Response 1: [We have defined the acronyms.] Thank you for pointing this out. We agree with this comment. Therefore, we have defined the acronyms such as gyrB and VOCs in Line 15, Page 1, potato dextrose agar (PDA) plate in Line 92, Page 3. |
|
· Comments 2: [An instance of “Fusarium” is left unitalicized in line 49.] |
|
Response 2: [We have revised it in line 49.] Thank you for pointing this out. We agree with this comment. Therefore, we have left the word “Fusarium” to be unitalicized in line 49. |
|
Comments 3: [Overall, the Methods section feels underdeveloped and should contain more technical information. In section 2.2 from the Methods, what the “antagonist” bacteria is is unclear. ] |
|
Response 3: [We have revised and marked it in red] Thank you for pointing this out. We agree with this comment. Therefore, we have added the content what anatgonist bacteria is in line 87, Page 2. The antagonistic bacteria exhibited good inhibition effect on Fusarium oxysporum. |
|
Comments 4: [The sentences surrounding line 90 are unclear. ] |
|
[We have rewritten this sentence in line 90.] Thank you for pointing this out. We agree with this comment. We have rewritten this sentence to make it clear. The sentence were revised as: A mycelial disc (6-mm diameter) of freshly 7-day-old culture was transferred on the center of 20 mL potato dextrose agar (PDA) medium plate. 100 μL of the bacterial inoculum with 1×106 colony forming units (CFU)/mLwas inoculated on the another plate with 15 mL of NA medium. Then, the two plates were rapidly sealed with polyethylene (PE) film. |
|
Comments 5: [The method of measurement used to supply data for the formula written in line 97 should be explained.] |
|
[We have added the measurement method in line 97.] Thank you for pointing this out. We agree with this comment. We have rewritten this sentence to make it clear. The mycelial growth was evaluated in two cross directions. |
|
Comments 6: [It is unclear in section 2.6 if full plants are being used for this experiment or just roots severed from a plant.] |
|
[We have used the roots to conduct this experiment in section 2.6.] Thank you for pointing this out. We agree with this comment. The biological efficiency of the VOCs to A. membranaceus root rot was conducted using the method as described by Zhang et al. (2020) and Wang et al. (2022). |
|
Comments 7: [A more comprehensive explanation of statistical analysis should be given for the data collection than the one provided in section 2.9. statistics used need to be made clear in figure caption.] |
|
[We have added the Shapiro–Wilk test for normality assessment.] Thank you for pointing this out. We agree with this comment. Each experiment was repeated three times, and the data were expressed as mean ± standard deviation (SD). For normality assessment, the experimental data were evaluated using the Shapiro–Wilk test. Statistically significant difference was determined by one-way analysis of variance (ANOVA) via Duncan’s multiple range test at P < 0.05 using SPSS version 19.0 software. |
|
· Comments 8: [In Figure 1a the differentiating markers (dot/box) between the control and treatment points are indistinguishable from one another.] · [We have changed the line connected with treatment points to dotted line] Thank you for pointing this out. We agree with this comment. To differentiate the markers between the control and treatment point, we used dotted line for treatments in Figure1a. |
|
|
|
· Comments 9: [In line 237 it is reported that disease incidence “significantly decreased by 13.33%,” but the accompanying figure appears to show that disease incidence has actually decreased to 13.33%; this is most likely a grammatical error but the difference between 13.33% and 86.67% is major.] |
|
[We have changed the sentence. ] Thank you for pointing this out. We agree with this comment. We are very sorry for our mistake. The disease incidence has actually decreased to 13.33%. |
|
· Comments 10: [I believe that Figure 5a is mislabeled on the 7 day treatment, both the control and treatment boxes are labeled “a” but I believe that they are actually statistically different from one another based on the rest of Figure 5.] |
|
[We have checked the analysis data, and revised the statistically different analysis from “a” to “b”.] Thank you for pointing this out. We agree with this comment. We are very sorry for our mistake. We have revised it in Figure 5. |
|
4. Response to Comments on the Quality of English Language |
|
Point 1: The English could be improved to more clearly express the research. |
|
Response 1: We are very sorry for the quality of English language. We have revised this manuscript carefully. |
|
5. Additional clarifications |
|
[We would like to thank you and the reviewers for the positive and valuable comments concerning our manuscript. We have checked it carefully. However, we are very sorry for our mistakes. Firstly, in Line 274 and Line 279, Paragraph 2, Page 9, Figure 6A and Figure 6A was changed to Figure 6a and Figure 6b. Secondly, we omitted the section of 3.7.3 GO and KEGG annotations analyses, and added this section in Line 279, Paragraph 2, Page 9. ] |

Reviewer 3 Report
Comments and Suggestions for Authors
The article addresses a current and important problem in the field of phytopathology and biological plant protection: the control of root rot caused by Fusarium oxysporum in Astragalus membranaceus – a medicinal plant of great economic importance. The isolation of the bacterial strain, its molecular identification, analysis of the antifungal activity of volatile metabolites (VOCs), their effect on the structure of the mycelium and transcriptomic changes in the pathogen are presented. The study concerns a real problem in the cultivation of medicinal plants. The isolation and identification of the biocontrol strain (B. paralicheniformis 2-12) were confirmed by two molecular markers (16S rRNA and gyrB), which increases the reliability of the identification. VOCs showed activity not only against F. oxysporum, but also against 15 other plant pathogens, which suggests a wide application potential. The use of SEM allows for precise documentation of mycelium damage. Significant changes in gene expression in response to VOCs have been demonstrated, which is an important contribution to understanding the mechanisms of action. Figures 6 and 7 are not very legible. The statistical analysis fragment contains basic information on how to conduct the statistical analysis, but requires stylistic improvement and supplementation with information on verifying the assumptions of the analysis of variance. In addition, it indicates the method and tool of analysis well. There is no information on whether the test assumptions, i.e. normality of the data distribution and homogeneity of variance (e.g. Shapiro-Wilk test, Levene's test), were checked before performing ANOVA. Line 190 - "Statistically significant difference was conducted" is grammatically and stylistically incorrect. It is better to use the phrase: "Statistical significance was determined" or "Significant differences were assessed". The discussion is written correctly. References contain 47 literature items.
Author Response
|
Comments 1: [Figures 6 and 7 are not very legible.] |
|
Response 1: [We have changed the pixel of each figures to be 600 dpi in Figures 6 and 7.] Thank you for pointing this out. We agree with this comment. Therefore, we have changed the pixel of each figures to be 600 dpi in Figures 6 and 7 in Line 289, Page 9 and Line 300, Page 10. |
|
Comments 2: [The statistical analysis fragment contains basic information on how to conduct the statistical analysis, but requires stylistic improvement and supplementation with information on verifying the assumptions of the analysis of variance. There is no information on whether the test assumptions, i.e. normality of the data distribution and homogeneity of variance (e.g. Shapiro-Wilk test, Levene's test), were checked before performing ANOVA.] |
|
Response 2: [We have added the the assumptions of the analysis of variance in Table 1.] Thank you for pointing this out. We agree with this comment. Therefore, we have added the parts of Shapiro-Wilk test in the Section of “2.9 Data analysis”, and added the Shapiro-Wilk p values in red in Table 1. |
|
Comments 3: [Line 190 - "Statistically significant difference was conducted" is grammatically and stylistically incorrect. It is better to use the phrase: "Statistical significance was determined" or "Significant differences were assessed". ] |
|
Response 3: [We have revised and marked it in red] Thank you for pointing this out. We agree with this comment. Therefore, we have revised it in red in Line 198, Paragraph 2, Page 5. |
